# Evaluating the Credibility and Reliability of Online Information on Cannabidiol (CBD) for Epilepsy Treatment in Poland

**DOI:** 10.3390/healthcare12080830

**Published:** 2024-04-14

**Authors:** Dawid M. Zakrzewski, Patrycja Podlejska, Wiktoria Kubziakowska, Kamil Dzwilewski, Przemysław M. Waszak, Marta Zawadzka, Maria Mazurkiewicz-Bełdzińska

**Affiliations:** 1Department of Developmental Neurology, Medical University of Gdansk, 80-210 Gdansk, Poland; dawid_zakrzewski@gumed.edu.pl (D.M.Z.); kubziakowska_49@gumed.edu.pl (W.K.); kamil.dzwilewski@gmail.com (K.D.); marta.zawadzka@gumed.edu.pl (M.Z.); mmazur@gumed.edu.pl (M.M.-B.); 2Department of Hygiene and Epidemiology, Medical University of Gdansk, 80-210 Gdansk, Poland

**Keywords:** CBD, epilepsy, social media, Epidiolex, cannabidiol, Internet

## Abstract

The interest in the potential therapeutic use of cannabis, especially cannabidiol (CBD), has increased significantly in recent years. On the Internet, users can find lots of articles devoted to its medical features such as reducing seizure activity in epilepsy. The aim of our work was to evaluate the information contained on the websites, including social media, in terms of the credibility and the reliability of current knowledge about the usage of products containing cannabidiol in epilepsy treatment. We used online available links found using the Newspointtool. The initial database included 38,367 texts, but after applying the inclusion and exclusion criteria, 314 texts were taken into consideration. Analysis was performed using the DISCERN scale and the set of questions created by the authors. In the final assessment, we observed that most of the texts (58.9%) were characterized by a very poor level of reliability and the average DISCERN score was 26.97 points. Additionally, considering the form of the text, the highest average score (35.73) came from entries on blog portals, whereas the lowest average score (18.33) came from comments and online discussion forums. Moreover, most of the texts do not contain key information regarding the indications, safety, desired effects, and side effects of CBD therapy. The study highlights the need for healthcare professionals to guide patients towards reliable sources of information and cautions against the use of unverified online materials, especially as the only FDA-approved CBD medication, Epidiolex, differs significantly from over-the-counter CBD products.

## 1. Introduction

Epilepsy is one of the most common neurological disorders, affecting around 70 million people worldwide [1]. By definition, it is a chronic disease characterized by the recurrence of unprovoked seizures, which result from excessive electrical discharges in the brain [2]. According to the International League Against Epilepsy (ILAE), the diagnosis of epilepsy includes not only the occurrence of at least two unprovoked seizures with intervals exceeding 24 h, but also a single unprovoked seizure and a probability of further seizures comparable to the general risk of recurrence after two unprovoked seizures, either within the next 10 years or upon diagnosis of an epilepsy syndrome [3]. Seizure types can be differential and vary from focal to generalized with different manifestation [4]. Furthermore, there are various types of epilepsy and epilepsy syndromes, classified by distinct scales such as the ILAE multi-level classification, which identifies over 20 epilepsy syndromes. Each syndrome is defined by a specific combination of clinical features, signs, symptoms, and electrographic patterns [5]. As regards the etiology of epilepsy, it is still unknown in about 50% of cases globally. The identified causes can be divided into the following categories: structural, genetic, infectious, metabolic, and immune [6].

The main treatment of epilepsy is pharmacological treatment. The choice of the most-suitable anti-epileptic drug depends on many dimensions such as the type of epilepsy, seizure type, or the patient’s characteristics [7]. However, about one-third of patients with epilepsy have seizures refractory to pharmacotherapy, then the disease can be termed “drug-resistant epilepsy” [8]. For these patients, new therapeutic strategies, both pharmacological and non-pharmacological, are subjects of current interest and their aims are ameliorating the symptoms of patients and answering the challenges of this disease [9].

Recently, the interest in the potential therapeutic use of cannabis in medicine has increased significantly [10]. The special attention of consumers is directed towards cannabidiol (CBD) [11]. On the Internet, users ascribe lots of therapeutic features to products containing CBD, among which is a reduction in seizure activity [12].

Experimental research has shown that CBD reduces inflammation, protects against neuronal loss, normalizes neurogenesis, and acts as an antioxidant [13]. These actions appear to be due to the multimodal mechanism of action of CBD in the brain. However, the mechanisms underlying its anticonvulsant and neuroprotective effects remain unclear [14].

Even though CBD is available in many different forms, such as oils, dried cannabis, dietary supplements, cosmetics, and many others, it is crucial to acknowledge that the first and only medicine containing cannabidiol approved by The United States Food and Drug Administration (FDA) is Epidiolex, whilst other unregulated formulations containing lower concentrations of CBD only have anecdotal evidence in support of their efficacy. The indication of this drug is the treatment of seizures associated with two rare and severe forms of epilepsy—the Lennox–Gastaut syndrome and the Dravet syndrome and with rare genetic disease—tuberous sclerosis complex (TSC), in patients one year of age and older. All of these conditions have their own specific characteristics. The Lennox–Gastaut syndrome is characterized by several seizure types that typically manifest before the age of 8. Additionally, there is severe cognitive impairment and an abnormal electroencephalographic pattern of slow spike-and-wave complexes [15]. The Dravet syndrome is the most well-known example of epilepsy caused by genetic factors, with over 80% of patients having a pathogenic variant of the SCN1A gene [16]. Similarly to the Lennox–Gastaut syndrome, seizures usually begin in childhood, but often much earlier, before the age of 20 months. Additionally, patients typically experience various types of seizures, with many being highly drug-resistant [17]. Tuberous sclerosis complex is a condition caused by variations in the TSC1 and/or TSC2 genes and is characterized by the presence of benign hamartomas in multiple organs and systems. Epilepsy affects the majority of patients with TSC and often begins during infancy. More than 60% of patients have drug-resistant epilepsy with focal seizures, infantile spasms in infancy, and various other types of seizures [18]. The efficacy and safety of Epidiolex has been thoroughly evaluated in clinical trials including randomized, double-blind, placebo-controlled clinical trials involving patients with either the Lennox–Gastaut syndrome, Dravet syndrome or tuberous sclerosis complex [19]. Among the patients with Lennox–Gastaut syndrome, there was a 37.2% (dose: 10 mg/kg/day) and 41.9% (dose: 20 mg/kg/day) monthly reduction in drop seizures over the 14-week treatment period, compared to a 17.2% reduction for those taking placebo [15], whereas among the patients with Dravet syndrome, there was a 38.9% reduction in monthly convulsive-seizure (dose: 20 mg/kg/day) over the 14-week treatment period, compared to a 13.3% reduction for those taking placebo [16]. In the case of seizures associated with TSC, clinical studies showed a 48.6% (dose: 25 mg/kg/day) reduction over the 16-week treatment period, compared to a 26.5% reduction for patients receiving placebo [18].

It is important to emphasize that, for other medical conditions, cannabidiol or other cannabis compounds should not be used without the context of a clinical trial. Similarly, the self-administration of smoked marijuana is not advisable. This is crucial information because, like other substances, cannabinol is not without adverse effects such as diarrhea, fatigue, vomiting, drowsiness, somnolence, changes in appetite and hepatic abnormalities [20]. Additionally, cannabidiol is a potent inhibitor of the CYP isoforms, which are responsible for metabolizing clobazam and many other antiseizure medications, including phenobarbital, phenytoin, carbamazepine, tiagabine, and valproate. These effects may partially explain both the toxicity and efficacy (by increasing responder rates) of cannabidiol in patients who are concurrently taking clobazam and other medications [21]. Importantly, the long-term safety of cannabidiol has not been established, and significant concerns exist regarding the potential negative effects of chronic cannabis use on brain development, cognitive function, and academic performance, especially in children with drug-resistant epilepsy, who may be more susceptible to such effects.

Nowadays, we are able to look up many online articles, statements, comments, movies, etc., that are concerned with epilepsy treatment by the use of products containing CBD [22]. The purpose of our work was to evaluate public information in terms of the credibility and the reliability of current knowledge about the usage of products containing cannabidiol in epilepsy treatment.

## 2. Methods

### 2.1. Search Strategy and Data Collection

A set of texts available online was used in this study. All were found using Newspoint sp.z.o.o. (Poland)—one of commercially available Internet content monitoring tools. The software allowed us to extract data from portals, microblogs, videos, social media, and online forums. To identify any data about epilepsy, we created a database using the search terms “padacz(epile)*” OR “epile*”. The “epile*” operator allowed us to use search terms including “epile” as epilepsy, epileptic, and epilepsja. The “padacz*” operator is the Polish language equivalent for the “epile*” operator.

Data export was carried out on 27 November 2021. Data were obtained for the period of 365 days (from 27 November 2020 to 27 November 2021). Every entry included the text, link, number of likes, follows, reach, and views. The initial database included 38,367 links (research data are provided within the Appendix A). The mentioned texts came from multiple social networking sites (Facebook, Twitter, etc.) and websites (journalistic and informational sites). If the source was a video platform, the video was viewed and treated as a text, which was finally evaluated in this study. 

The data concern both the treatment of adults and children.

The quoted online academic publications were not included in this study.

All research data were publicly available, so no bioethics committee permission was necessary.

### 2.2. Inclusion and Exclusion Criteria

From the mentioned group, there were 38,367 texts. The inclusion criteria were mentioned for CBD therapy. For that, we searched terms including “CBD”, “kannabidiol”, and “cannabidiol” among the initial database. After that, a group of texts was created consisting of 1126 links. 

The exclusion criteria included whether the data were duplicated on the same platform, hence excluding 617 files. Numerous duplicates were related to the multiple sharing of a given text by one platform while keeping the original source link, and thus, all statistics. A total of 509 files were retained for analysis. To the study, the 314 of 509 texts were used. The other reasons for exclusion are contained in Table 1.

### 2.3. Scoring System

The collected texts were analyzed in terms of reliability and checked to determine whether they corresponded to current medical knowledge through questions, which were created by authors, concerning the efficiency of epilepsy treatment, predicted effects, indications, safeness, and side effects of CBD therapy. The questions asked during the analysis by the authors concerned the key issues needed to understand the topic and illustrate its entirety. They made it possible to check the correctness of the text and its compliance with current medical knowledge, and above all, they answered the question of whether a given text addresses a specific sector of knowledge that is part of the whole issue. Thereafter, the articles were assessed using the DISCERN (Quality Criteria for Consumer Health Information) scale. The DISCERN system is a validated instrument for assessing the reliability of medical information, created at the University of Oxford, Division of Public Health and Primary Health Care, at the Institute of Health Sciences [23]. It includes 16 questions about various key treatments contained in the publications, which are included in Table 2. For each question, the text was scored from 1 to 5 points (total sum from 16 to 80). By this subjective point-scoring, the authors vaulted each text from 1 point, which meant a definite “no”, to 5 points, which meant a definite “yes”, whereas a score from 2 to 4 points was given if the authors felt that the text met the criterion in the question to an incomplete extent. Question 16 was a subjective assessment of the assessor of the credibility and reliability of a given text obtained after analyzing all the previous questions. The Discern scale and its general statement are available online and well labeled on Table 2.

### 2.4. Statistical Methods

Raw data were collected in Excel spreadsheets (Microsoft^®^ Excel^®^ for Microsoft 365 MSO—version 2403 build 16.0.17425.20124, Microsoft, Redmond, WA, USA). Statistical analysis was performed using STATISTICA 10.0 software (StatSoft Inc., Tulsa, OK, USA). All of the quantitative variables were tested using the Kolmogorov–Smirnov test to meet the criteria of a normal distribution (Gaussian distribution). Depending on whether the variables met the normality condition, appropriate statistical tests were applied at further stages. Continuous data were presented as medians and quartiles if they did not meet the conditions of the normal distribution. For comparisons between two groups, the parametric *t*-test or nonparametric Mann–Whitney U test were used. For the comparison between multiple groups, an analysis of variance (for the variables of parametric distribution) or the Kruskal–Wallis test (variables of non-parametric distribution) were used. For comparing qualitative survey data, Pearson’s chi-square test (with appropriate Yates’ Correction for small observed frequencies) was used. The cut-off level was set at *p* < 0.05 for statistical significance.

## 3. Results

### 3.1. The Database

The collected texts came from multiple sources, such as social networking, journalistic and informational sites. The most common source in this study were social media (Facebook, You Tube and Twitter). Table 3 shows the collective counts of all sources used in the study.

Then, texts were divided into many categories, depended on the form of the text, showed in Table 4. All of them were analyzed both in the categories and collectively.

### 3.2. Quality Analysis in General

In the final assessment, most of the texts (58.9%) were characterized by a very poor level of reliability, being at the lowest level of the DISCERN scale, gaining 16–26 points. The ration to one of the quality category can be found of Figure 1. The lowest-tared text gained 16 points. None of the texts were classified as very good quality material, the best-rated text gained 59 points. The average DISCERN score was 26.97 points. The best rated questions were question no. 1 (2.96) and question no. 2 (2.53), regarding the assumptions and goals of the texts. Questions no. 11 (1.16) and question no. 12 (1.04) concerning the risk of treatment and the consequences of not treating, respectively, received the lowest scores. The rating of all the DISCERN questions are presented in Table 5.

### 3.3. Quality Analysis Depending on a Form of a Text

The analysis was also undertaken for each category of the form of the text. Most of the texts were classified as a post from an informational site—199 texts. The mean point score of the DISCERN scale in that category was 27.18. The highest average score (35.73) corresponded to entries on blog portals. The lowest average score (18.33) came from comments and form online discussion forums (Figure 2). The difference between the individual categories is visible not only in the average score, but also in the comparison of a given category between individual questions. The most noticeable differences between the answers to the given questions of the DISCERN scale were noticed in questions 6, 8, 10, and 13. Higher scores for these questions were recorded for video and blog sources, while the lowest scores were for discussion forum entries and comments (Table 6).

### 3.4. Media Reach

During the analysis of texts, statistical data on the reach, views, and shares of a given material were also collected. The basic statistics of interactions are included in Table 7. When analyzing the given data between the created categories, it was not possible to statistically determine significant differences due to the missing information about interactions in some groups. Videos, posts from social media, and news portals were characterized by the largest number of media interactions.

### 3.5. Author’s Questions Analysis

Although each of the texts included in this study was related to epilepsy and CBD, most entries lacked key information regarding the indications, safety, desired effects, and side effects of CBD therapy for epilepsy. On average, about 80.31% (58.60%:99.68%) of the texts did not contain answers to the questions asked by the authors. The questions that missed the most entries concerned standard epilepsy treatment protocols (question No. 1 and 3) and the most information could be found on CBD indications (Q No. 4), CBD anti-seizure effect (Q No. 7), and CBD safety (Q No. 10 and 11). However, none of these issues were raised by at least half of the authors of the surveyed texts. Moreover, the analysis found numerous expressions misleading the reader, which are not consistent with current medical knowledge. The greatest number of errors concerned the indications for CBD therapy, and as much as 10.19% of the texts contained information that CBD therapy is indicated in all types of epileptic seizures. An equally high level of non-compliance concerned side effects, where 9.55% of the texts considered the lack of possibility of their occurrence during CBD therapy. The third most frequently repeated myth, in 4.78% of texts, is the possibility of completely curing epilepsy using CBD. In the remaining questions, incorrect information occurred in up to 3% of the recorded answers.

During analyzing the texts, there were also two open questions regarding the side effects and the form of CBD taken.

The most frequently mentioned side effects (among texts that considered the occurrence of these effects as possible) during CBD therapy are sleep problems (6.37%), gastrointestinal problems (5.73%), appetite changes (2.55%), pressure changes (2.55%), dry mouth (2.23%), and allergic reactions (1.91%).

The above information concerned the use of CBD as a compound and its form was not clearly identified. That is why data on the form of the preparation used was also collected. The studied sources most often mentioned CBD in the form of oil (219), medicine (55), and dried leaves (35), and others were mentioned less often, like candies, cosmetics, creams, drinks, flour, and tea.

The set of questions contained in the formula, with the percentages of answers, can be found in Table 8.

## 4. Discussion

### 4.1. The Current Study

With the growing interest in CBD in the world of science, there is also an increase in interest in it on the open market, its popularity in social media, and the curiosity of many patients regarding whether it may be a potential treatment for them. That is why we decided to analyze the quality of information available on the Internet about the impact of CBD on the treatment of epilepsy, determining what information patients looking for answers there may encounter. It turned out that most of the texts (58.9%) were characterized by a very poor level of reliability, and the average DISCERN score was 26.97 points, confirming that the reader was left with unreliable information about epilepsy therapy using CBD. Such a weak conclusion resulted from the reader being misled more than once or omitting facts important for understanding the topic, most often advertising the use and purchase of CBD-containing products. The texts lacked information about the most important issues regarding CBD therapy, as well as standard protocols for pharmacological treatment in epilepsy. Most incorrect data concerned indications for therapy, side effects, and prognosis for complete recovery thanks to therapy. It can be assumed that most of this information is intended to encourage the consumer to purchase CBD products, not to inform them about the full issue of its use in epilepsy.

The texts were separately analyzed in the category of the form of the text. Is there a difference between the subgroups? The best result was obtained by materials from blogs and videos, and the worst from comments and discussion forums. 

### 4.2. Quality of Internet Source in Other Works

The Internet has become one of the most important sources of information that people reach for. That is why it is extremely important to monitor the quality of content posted in it. Unfortunately, it has been shown that 40% of the content related to medicine in Polish is not true [24], but in the case of various topics, such as our study, this percentage may be even higher.

The medical literature on the quality of online information about CBD for epilepsy is still sparse, which is why we have decided to include this discussion as part of the ongoing debate about the quality of medical information available online.

Silek et al. [25] conducted a study similar to ours, which analyzed 100 films using the DISCERN scale and the GQS scale, concluding that only 25% of the materials are of good quality, and as much as 43% of average quality and 32% of poor quality. These results are comparable to those from our study. However, they present a higher percentage of good quality items. Nevertheless, the data used in this study come only from videos from the YouTube platform, which correlates with our results, in which videos were among the best-rated sources of information.

Unfortunately, YouTube is not a perfect source of information, and the search results for some medical topics are of low quality. This was the case with the work of Salah et al. [26], which verified the accuracy and quality of the content of 30 videos related to vitiligo. The average result of the data used in the DISCERN scale was rated at 30.5 points, and most of the articles were classified as poor quality. In this study, the GQS and ANDI scales were additionally used, which confirmed the conclusions of the DISCERN scale. In our study, despite not including both scales, the authors’ questions were used, which helped to fully analyze what information was missing and which issues were incorrect.

The same method of analysis was used by Tkaczuk et al. [27] to examine the quality of information on the Internet about Onasemnogen Abapervovak gene therapy in multiple sclerosis. After analyzing the DISCERN scale and the author’s questions, the authors came to the conclusion that most of the texts had a poor (48.65%) reliability level and the mean DISCERN score was 39.66 points, which indicates that the online material on the gene therapy is of “medium” quality. However, the recipients remained with an incomplete picture of the issue due to significant gaps in key information needed to understand the topic, which was also noticed in the analysis of our study. Both studies were conducted on a database composed of many news, journalistic, and community websites, presenting the issue in a broad sense of the “Internet”. Jayasinghe et al. [28] constructed their study in a similar way, evaluating the first 100 websites on the subject of COVID-19 on Yahoo!, Google, and Bing search engines. The validated DISCERN scale, LIDO scale, and FRES scale were used for the assessment. The majority of websites on COVID-19 for the public had moderate-to-low scores with regard to readability, usability, reliability, and quality.

An even worse quality result, in relation to ours and those indicated above, was the result of the study conducted by Johnson et al. [29]. It took into account the 200 most popular websites (selected using web-scraping software—Buzzsumo.com) on the four most common cancers and assessed the quality of information posted there. Also, in this study, the articles came from various social media—Facebook, Twitter, Pinterest, etc. What made this work different from ours was, among other things, the scales used. The authors used the Likert scale and the Wilcoxon test. The Likert scale includes five statements from complete rejection to complete acceptance, while the Wilcoxon test was designed to check the statistical relationship between total engagement and engagement on Facebook between harmful information and disinformation. It found that a third of the websites surveyed in 2018 and 2019 contained misinformation, and most even potential harm. This is a much larger percentage of information inconsistent with EBM than what we noticed in the evaluation of CBD in the treatment of epilepsy, which strongly tilts the scale of the discussion regarding the quality of information on the web to the side of disinformation on many portals.

Many authors, however, prefer to focus on a narrow goal by analyzing materials on a specific Internet platform. This is how Turner et al. [30] constructed their work, thus evaluating the issues raised on Twitter regarding cannabidiol and classifying them according to the mood of reception into positive negative and neutral. This division was made possible by the VADER model. The study showed similarities and differences in terms used in private and commercial tweets, thus providing information on the degree of interest in individual CBD-related terms. In both, the topics of pain, sleep, and anxiety were the most frequently discussed. Commercial tweets were more about nutrition and fitness, and personal tweets were about autism and alcohol. These are issues that often accompanied the texts we studied. However, they were not analyzed in this respect, due to the lack of translation of these issues directly into the treatment of epilepsy. The study also suggests that CBD is generally viewed favorably for its medical uses. But where does this rating come from? This question is indirectly answered by Soleymanpour et al. [31] in his work searching and evaluating popular therapeutic claims about CBD contained in tweets. In these tweets, marketing messages were identified using the SVM and LR classifiers, and therapeutic messages using created patterns. This study confirmed that pain, sleep, and anxiety are the topics most frequently mentioned in CBD messages. More than half of the tweets were classified as marketing communications, and it was noted that the marketing claims are clearly inconsistent with FDA guidelines. This suggests an attempt to manipulate information and deliberately present CBD in a better light by persuading the recipient to buy CBD food products and oils.

The credibility and quality of medical information presented by cannabis sellers decided to check Ng et al. [32]. He analyzed, using the DISCERN scale, the first 33 websites of cannabis retailers found for the Canadian market. The obtained results confirmed the hypothesis put forward by the authors about the low quality of information provided by sellers. The mean DISCERN score was 36.83 (SD = 9.73). As in our study, this low score can be attributed to, among other things, not addressing the uncertainties in the scientific evidence about cannabis, not making references to the medical literature, or providing additional sources of support, and not showing the impact of cannabis use on quality of life, alternatives for its use, the risks of its consumption, and the consequences of not taking any treatment. In our study, the result of the quality analysis was even lower, which may suggest that the inclusion of various websites, articles, and comments in the study, as well as narrowing down the search to the use of cannabis in particular disease entities, additionally reduces the quality and reliability of data available on the Internet.

### 4.3. Limitations and Advantages

One of the most important limitations of the study is the dynamic of the rapidly developing Internet and emerging websites containing materials related to the use of CBD in the treatment of epilepsy. We presented here only a fragment of the total information available on social networks in the indicated period of time. Furthermore, the research takes into account only Polish materials and does not consider information written or presented in other languages. 

The published content was restricted to very general, fragmentary information, which meant that the recipient could obtain significantly limited knowledge. The subject of CBD use in epilepsy treatment is a complex concept and requires, in our opinion, a detailed presentation of its issues and the possibility of treating epileptic patients in Poland. 

The questions we created, according to the authors, defined the completeness of the issue of products containing CBD in epilepsy. Unfortunately, these questions are not validated.

In addition, it should be noted that some of the statements we studied were only a few sentences long, so it was difficult to assess such entries on the DISCERN scale. Many of these articles were sponsored by the manufacturers of products containing CBD. For this reason, they may not be objective, and in many cases, one could even get the impression that the content was not intended to educate readers, but only to show oils and other products in the best possible way.

Nevertheless, the great advantage of our study is the evaluation of articles from various social media, including from Facebook, YouTube, Pinterest, and more. The analyzed materials came from many databases, which makes the study more reliable. The DISCERN scale we used has been assessed by more than one researcher, which makes it more impartial and objective.

### 4.4. Future Direction

We strongly believe that our study, which is not devoid of limitations, opens the way for future studies on the reliability of public information and its compliance with medical knowledge.

In our research, we only analyzed Polish texts about the treatment of epilepsy with CBD. Hence, in the future, it would be worth focusing on the subject in a wider range. We are convinced that checking English data can expand this study significantly. 

Moreover, there are some differences between texts describing the user’s own experience and commercial texts that we noticed during our analysis. Thus, in future studies, the authors may take the character of information into consideration to compare its quality and reliability.

In addition, it is important to mention that most of evaluated materials discuss commercially used CBD products, not Epidiolex, which is the only FDA-approved medication containing CBD. Until the end of year 2023, Epidiolex was not refundable in Poland. However, since the beginning of year 2024, Epidiolex is refundable in Poland for patients over 2 years old in treatment of Dravet syndrome Lennox–Gastaut syndrome and tuberous sclerosis in combination with Clobazam. We suspect that this innovation may lead to some changes in the information on CBD treatment available on social media. Then, the future studies may be essential to verify our results and conclusions. 

Our work shows the value of evaluating information commonly available to the patient on the Internet, which indicates the need for further studies on the compatibility of online data about the treatment of diseases other than epilepsy with EBM and thus, the credibility of the knowledge that patients are able to acquire on their own.

## 5. Conclusions

The popularity of CBD has been constantly growing in recent years, and with it the interest in its antiepileptic effects among patients. This study evaluates the quality of internet information about CBD therapy in epilepsy. The DISCERN instrument was used to evaluate 314 texts. Then, texts were also analyzed, meeting the eligibility criteria from authors questions. Our findings indicate that the quality of knowledge about CBD therapy in the Internet is poor and has a lot of shortcomings, which is really important to remember when reading social media posts and sites from CBD vendors. It may also be of value to replicate this study across other jurisdictions and assess the accuracy of information provided online in the future. We strongly believe that our efforts demonstrate that research on social data is significant to surveil society’s attitude and individual patients’ possibility to encounter reliable information that is compliant with medical standards. In conclusion, it is worth underlining that healthcare professionals should warn patients against using unverified sources of information, while pointing to reliable materials.

Finally, it is worth emphasizing the difference between Epidiolex—highly purified CBD (currently available in Poland only as Rescue Access to Drug Technologies (RDTL))—and other over-the-counter preparations containing CBD on the market. The latter are most often untested dietary supplements, containing small amounts CBD, significantly lower than in the Epidiolex drug. Currently, patients, guided by online recommendations, reach for these products without consulting a doctor, often giving up another form of treatment. They believe, based on Internet sources, that these supplements are the most effective form of treatment epilepsy, without side effects. Unfortunately, both of the above statements are untrue.

## Figures and Tables

**Figure 1 healthcare-12-00830-f001:**
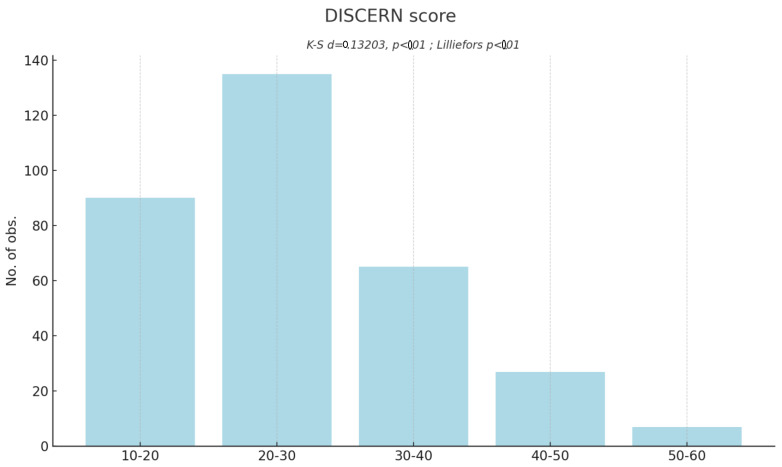
Histogram. Total DISCERN score account.

**Figure 2 healthcare-12-00830-f002:**
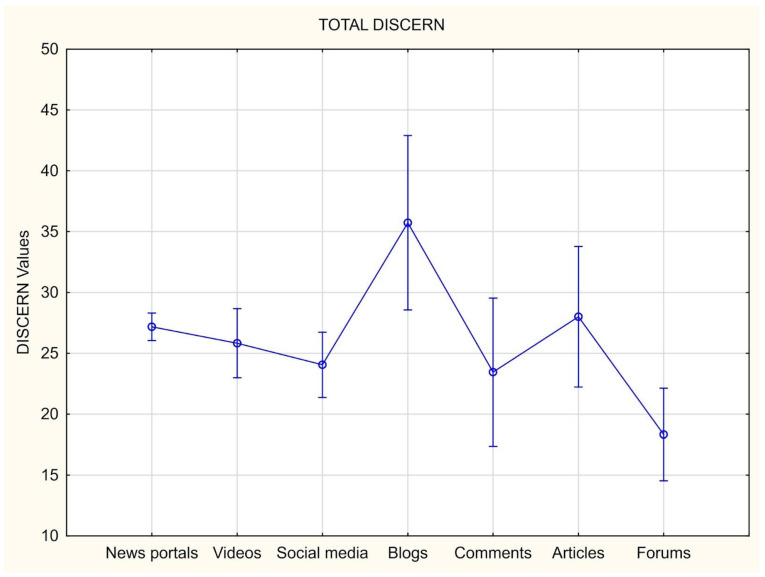
Total DISCERN score account depending on a form of a text.

**Table 1 healthcare-12-00830-t001:** The reasons for exclusion.

Reason for Exclusion	Number of Excluded Texts
Duplication on the same platforms	617
Excessively short text (<3 sentences)	70
Text only about animals	58
CBD and epilepsy mentioned in the text, but not in the contexts of each others	34
CBD only as an add	13
Inaccessible movie	5
Keywords only in comment behind the text	5
Epilepsy in another link on this website	3
Language other than polish	3
Keywords only as hashtags	2
Text found out as a table of contents	2

**Table 2 healthcare-12-00830-t002:** DISCERN instrument variables.

No. of the Question	The Question
1	Are the aims clear?
2	Does it achieve its aims?
3	Is it relevant?
4	Is it clear what sources of information were used to compile the publication (other than the author or producer)?
5	Is it clear when the information used or reported in the publication was produced?
6	Is it balanced and unbiased?
7	Does it provide details of additional sources of support and information?
8	Does it refer to areas of uncertainty?
9	Does it describe how each treatment works?
10	Does it describe the benefits of each treatment?
11	Does it describe the risks of each treatment?
12	Does it describe what would happen if no treatment is used?
13	Does it describe how the treatment choices affect overall quality of life?
14	Is it clear that there may be more than one possible treatment choice?
15	Does it provide support for shared decision-making?
16	Based on the answers to all of the above questions, rate the overall quality of the publication as a source of information about treatment choices

Source: http://www.discern.org.uk/discern_instrument.php, accessed on 1 December 2022.

**Table 3 healthcare-12-00830-t003:** Data sources.

Portal	Count	Percent of Total
Facebook	37	11.78
YouTube	33	10.51
Onet	11	3.50
Hyperreal	8	2.55
Twitter	6	1.91
MSN Polska	5	1.59
Wykop	4	1.27
Głos Wielkopolski	3	0.96
Instagram	3	0.96
Bielsko.info	3	0.96
Gazeta	3	0.96
Wprost	3	0.96
Forum—Gazeta	3	0.96
Siedlecki Portal Informacyjny	3	0.96
Medonet	3	0.96
Hej Mielec	3	0.96
Bomega	3	0.96
Portal Pszczyński	3	0.96
Familie	2	0.64
Female	2	0.64
Gazeta Oławska	2	0.64
Uroda i zdrowie	2	0.64
Rzeczpospolita	2	0.64
Życie Pabianic	2	0.64
Together Magazyn	2	0.64
portEl	2	0.64
Portal Jaworzyński	2	0.64
dlaLejdis	2	0.64
Calisia	2	0.64
Dziennik Naukowy	2	0.64
NaTemat	2	0.64
Biuro prasowe	2	0.64
Ekologia	2	0.64
eKutno	2	0.64
Rybnik	2	0.64
oFeminin	2	0.64
Menopauza	2	0.64
Czecho—Wiadomości	2	0.64
Powiat Suski	2	0.64
Zephyrnet	2	0.64
ZW—Home	2	0.64
Dziennik Internautów	2	0.64
Webniusy	2	0.64
Regionalny Portal Informacyjny (Pruszków)	2	0.64
Fashion and Beauty	2	0.64
BFN PL Today	2	0.64
Kto Cię Wyleczy	2	0.64
Moja Cukrzyca	2	0.64
Forum Miasto Kobiet	2	0.64
Tygodnik Ostrołęcki	2	0.64
The other portals were a small percentage of total.		

**Table 4 healthcare-12-00830-t004:** Categories of texts used in the study.

Category	Frequency Table: Category of Text			
	Count	Cumulative Count	Percent	Cumulative Percent
News websites	199	199	63.3758	63.3758
Movies	58	257	18.4713	81.8471
Social media	20	277	6.3694	88.2166
Blogs	15	292	4.7770	92.9936
Comments	9	301	2.8662	95.8599
Articles	10	311	3.1847	99.0446
Forums	3	314	0.9554	100.0000
Missing	0	314	0.0000	100.0000

**Table 5 healthcare-12-00830-t005:** Mean discern score for single question.

Question of DISCERN Scale	Descriptive Statistics				
	ValidN	Mean	Min.	Max.	Std. Dev.
Are the aims clear?	314	2.9650	1.0000	5.0000	1.25961
2. Does it achieve its aims?	314	2.5338	1.0000	5.0000	1.21469
3. Is it relevant?	314	2.2701	1.0000	5.0000	1.19314
4. Is it clear what sources of information were used to compile the publication (other than the author or producer)?	314	1.7612	1.0000	5.0000	1.17629
5. Is it clear when the information used or reported in the publication was produced?	314	1.5287	1.0000	5.0000	1.05480
6. Is it balanced and unbiased?	314	1.8344	1.0000	5.0000	1.16305
7. Does it provide details of additional sources of support and information?	314	1.4682	1.0000	5.0000	0.98581
8. Does it refer to areas of uncertainty?	314	1.7293	1.0000	5.0000	1.08430
9. Does it describe how each treatment works?	314	1.3280	1.0000	4.0000	0.60142
10. Does it describe the benefits of each treatment?	314	1.5064	1.0000	4.0000	0.66996
11. Does it describe the risks of each treatment?	314	1.1656	1.0000	4.0000	0.49724
12. Does it describe what would happen if no treatment is used?	314	1.0414	1.0000	4.0000	0.29076
13. Does it describe how the treatment choices affect overall quality of life?	314	1.3153	1.0000	4.0000	0.60294
14. Is it clear that there may be more than one possible treatment choice?	314	1.4522	1.0000	5.0000	0.79894
15. Does it provide support for shared decision-making?	314	1.6210	1.0000	5.0000	1.11896
16. Based on the answers to all of the above questions, rate the over	314	1.4665	1.0000	4.0000	0.78833
Total DISCERN score	314	26.9713	16.0000	59.0000	9.02512

**Table 6 healthcare-12-00830-t006:** The differences of the single question results in subgroups.

Category	News Portals	Movies	Social Media	Blogs	Comments	Articles	Forums
Q.6 Is it balanced and unbiased? Means	1.8592	1.7759	1.3500	2.8667	1.4444	1.7000	1.0000
Q.8 Does it refer to areas of uncertainty? Means	1.7085	1.8276	1.2000	2.4000	1.3333	2.2000	1.0000
Q.10 Does it describe the benefits of each treatment? Means	1.4824	1.4310	1.6000	2.3333	1.0000	1.6000	1.0000
Q.13 Does it describe how the treatment choices affect overall quality of life? Means	1.2915	1.2413	1.1500	1.9333	1.2222	1.8000	1.0000

**Table 7 healthcare-12-00830-t007:** Basic social media statistics of the included material.

Variable	Descriptive Statistics				
	Valid N	Mean	Minimum	Maximum	Std. Dev.
Reach	287	27,645.23	4.0000	300,000.0	63,153.61
Views	17	28,680.82	7.0000	176,069.0	57,159.30
Shares	25	68.84	1.0000	1152.0	229.77

**Table 8 healthcare-12-00830-t008:** Information and statements found in the analyzed material.

	Count	Cumulative Count	Percent	Cumulative Percent
Category	1. Is epilepsy treatment reimbursed in Poland?			
Missing	313	313	99.6816	99.6816
Yes	1	314	0.3185	100.0000
Category	2. Is epilepsy treatment effective?			
Missing	298	298	94.9045	94.9045
No	13	311	4.1401	99.0446
Yes	3	314	0.9554	100.0000
Category	3. Does anti-epileptic treatment have many side effects?			
Missing	310	310	98.7261	98.7261
Yes	4	314	1.2739	100.0000
Category	4. Is CBD used in all types of epilepsy?			
No, only in certain types/teams	99	99	31.5287	31.5287
Missing	184	282	58.5987	89.8089
Yes	32	314	10.1911	100.0000
Category	5. Are there side effects with CBD therapy?			
Missing	239	239	76.1147	76.1147
Yes	45	284	14.3312	90.4459
No	30	314	9.5541	100.0000
Category	6. Is CBD therapy an individual therapy replacing other treatments?			
Missing	248	248	78.9809	78.9809
No	57	301	18.1529	97.1338
Yes	9	314	2.8662	100.000
Category	7. Does CBD therapy reduce the frequency of seizures?			
Yes	113	113	35.9873	35.9873
Missing	197	310	62.7389	98.7262
No	4	314	1.2738	100.000
Category	8. Does CBD therapy result in symptoms withdrawal/complete recovery (no seizures)?			
No	33	33	10.5096	10.5096
Missing	266	299	84.7134	95.2230
Yes	15	314	4.7770	100.0000
Category	9. Is long-term CBD treatment possible?			
Missing	255	255	81.2102	81.2102
Yes	58	313	18.4713	99.6815
No	1	314	0.3185	100.0000
Category	10. Is CBD treatment safe (does it not cause serious side effects)?			
Missing	217	217	69.1083	69.1083
Yes	93	310	29.6178	98.7261
No	4	314	1.2739	100.0000
Category	11. Does CBD affect cognitive function during therapy?			
Missing	192	192	61.1465	61.1465
Yes, positively	66	258	21.0191	82.1656
No	53	311	16.8790	99.0446
Yes, negatively	3	314	0.9554	100.0000
Category	12. Can you become addicted to CBD during treatment?			
Missing	224	224	71.3376	71.3376
No	86	310	27.3885	98.7261
Yes	4	314	1.2739	100.000
Category	13. Does CBD interact with other epilepsy medications?			
Missing	289	289	92.0382	92.0382
Yes	24	313	7.6433	99.6815
No	1	314	0.3185	100.0000
Category	14. Is the cost of CBD treatment high?			
Missing	282	282	89.8089	89.8089
No	7	289	2.2293	92.0382
Yes	25	314	7.9618	100.0000
Category	15. Is CBD treatment reimbursed in Poland?			
Missing	274	274	87.2612	87.2612
No	38	312	12.1019	99.3631
Yes	2	314	0.6369	100.0000
Category	16. Is the effectiveness of synthetic CBD comparable to plant-derived CBD?			
Missing	269	269	85.6688	85.6688
No	43	312	13.6943	99.3631
Yes	2	314	0.6369	100.0000

## Data Availability

The data that support the findings of this study are openly available via the Internet (see Section 2). Detailed databases used in this research are available from the corresponding author upon reasonable request.

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
