# Peer review of "Evaluating the Credibility and Reliability of Online Information on Cannabidiol (CBD) for Epilepsy Treatment in Poland"

_healthcare, 2024, doi:10.3390/healthcare12080830_

Round 1

Reviewer 1 Report

Comments and Suggestions for Authors

In this study, the authors evaluate the information available online (different platforms) regarding CBD use in epilepsy. They aim to evaluate the usefulness  and validity of available information present online. Overall, the organization of the material presented can be improved and could be more concise. My biggest critique is the applicability and usefulness of this information is questionable. The authors have included in their study materials all the information available in different forums including comments and it is not surprising that they find it unreliable and misleading. Also, out of all the collected information, only 314 texts were included and this essentially does not represent the information available online. It would make more sense to study the reliability  of information on specific educational sites or valid advertisement sites and to see their usefulness and accuracy as comments and discussions are mostly personal opinions and we can not rely on the source of information anyways.

Another concern regarding the study is the choice of medication. CBD in its various forms is used and is not FDA regulated. Marijuana is banned in some states and people likely use the terms (CBD and marijuana, other synthetic forms) and we do not know exactly the effects of which specific medicine they are reporting about. Specific information about Epidiolex if available and its effect on epilepsy can be studied in a more reliable way and will provide some more useful data.

Comments on the Quality of English Language

Can be improved. Minor concerns.

Very extensive, can be concise. Discussion and introduction sections are lengthy. In the discussion section, it will be worthwhile to only report the findings and comparison with the available data and limitations. 

Author Response

Dear reviewer,

Reviewer 2 Report

Comments and Suggestions for Authors

Dear authors,

Thanks for the submission. This article provided the high percentage of unverified and unreliable online medical resources needed for healthcare professionals to put more effort into guiding patients during and after clinical interventions.

However, some issues still need to be further addressed.

1: In 2.1 Search Strategy and Data collection section, should mention that the quoted online academic publications were not included in this study.

2: In 2.3 Scoring system line 119, should clarify what the scoring from 1-5 indicates,  since it’s very subjective.

3: Line 156, punctuation is not consistent.

4: Since the study was performed and only included the resources in Polish, this key information should be addressed in the title. 

Author Response

Dear Reviewer,
